# Species independence of eye lens dimensions in teleosts and elasmobranchs

**Rannveig Rögn Leifsdóttir** [ID] *, **Steven E. Campana** [ID]

Life and Environmental Science, University of Iceland, Reykjavik, Iceland

* rannveigrl@gmail.com

**Data Availability Statement:** All relevant data are within the paper and its Supporting information files.

**Funding:** The author received no specific funding for this work.

## Abstract

The vertebrate eye lens grows incrementally, adding layers of elongated, tightly packed lens fiber cells at the outer margin of the lens. With subsequent growth, previously-deposited fiber cells degrade, leaving a region of fully denucleated and organelle-free cells which are responsible for the high transparency and low light scattering characteristics of the lens. The objective of this study was to determine if the horizon separating the gelatinous outer cortex of the lens from its hardened interior occurred at a consistent location within the lens of several teleost and elasmobranch fish species, and could be linked to fiber cell morphology or function. A fixed ratio of 0.69±0.01 of hardened eye lens diameter (HD) to overall eye lens diameter (LD) was observed in a broad size range of Atlantic cod (*Gadus morhua*), haddock (*Melanogrammus aeglefinus*), thorny skate (*Amblyraja radiata*) and round ray (*Rajella fyllae*). The location of the hardened lens horizon was similar to that reported for optical plasticity and spherical aberration, but not that of fiber cell denucleation, suggesting that fiber cell dehydration continues after the loss of internal organelles. Our findings support a previous suggestion that the maintenance of optical quality during fish eye lens growth requires a precisely-fixed HD:LD ratio, while the ubiquity of a fixed ratio across fish taxa may suggest that many fish species possess a common refractive index profile. The linear relationship between HD and fish length should allow fish length to be backcalculated from the diameter of the isolated lens core, thus aiding research using isotope ratios of lens laminae or inner cores to reconstruct early life history events.

## Introduction

The vertebrate eye lens has received attention in recent years as a potential biochemical recorder and source of age information [1–3]. Radiocarbon dating and amino acid racemization rates have been applied to eye lenses as an age validation method for both marine mammals and fish species [2–5]. Eye lenses have also been utilized as a repository for isotope derived information on fish life history [1, 6]. Although eye lenses are layered, the layers are not themselves reliable age indicators, as is the case with calcified structures like otoliths [7]. Nevertheless, the absence of physiological activity in the solidified lens core gives it some properties in common with otoliths.

**Competing interests:** The authors have declared that no competing interests exist.

Unlike most organic tissues, the vertebrate eye lens grows incrementally, with much of the central region becoming metabolically inert sometime after deposition [8–10]. The innermost part of the lens is known to form during prenatal development and thus contains proteins synthesized around the time of birth [9, 11]. Later growth adds layers of elongated, tightly packed lens fiber cells, which inter-connect along both their short and long axes. The nuclei and other organelles of the fiber cells subsequently degrade, leaving a fully denucleated and organelle-free cell [12]. It is the absence of these intracellular structures in the lens that contributes to its high transparency and low light scattering characteristics. In the fish lens, all fiber cells within the inner 92% of the lens radius are fully denucleated and free of organelles [13]. The light focusing characteristics of the lens comes from a refractive gradient within the lens itself [12].

The typical fish eye lens consists of three morphologically-distinct regions. 1) The outer cortex of the lens is gelatinous, with its structural integrity maintained by an outer lens capsule [14]. 2) Inside of the lens cortex is a hard, nearly incompressible sphere of dense protein, termed the core by some authors [1, 14, 15] but not by others. The lens cortex and "core" are easily distinguished in fresh material, with the former having the consistency of gelatin. 3) Less easily distinguished is the embryonic region of the core, which is apparent as poorly ordered layers of fiber cells visible in sections [12]. This embryonic region can sometimes be isolated by peeling away layers of hardened lens until no further peeling is possible, but can also be approximated as the smallest possible central region of the hardened portion of the lens [2, 6]. This central region of the hard part of the lens, which can reasonably be inferred to represent the earliest life history of the fish, has variously been termed the lens core [6], nucleus [2] or the central core [1]. It is noteworthy that the "core" of Fernald and Wright (1983) [14] and others does not correspond with the "core" of Vecchio and Peebles (2020) [6] and others. To avoid confusion, we refer to the entire hardened portion of the lens medial to the gelatinous lens cortex as the HL (hardened lens) and the embryonic region as the CHL (central hardened lens).

Considerable attention has been given to the isotopic composition of the CHL, given that it represents a proxy for the early life history of the fish. Much less attention has been given to the HL as a whole, despite its important role in maintaining focused vision. Working with a single fish species, Fernald and Wright (1983) recorded a cortex-HL horizon at 67% of the lens radius, which was invariant across a range of fish sizes. In a different species of the same family, Schartau et al. (2009) [13] demonstrated that the horizon corresponding to complete absence of intracellular structures was fixed at 92% of the lens radius, but that a threshold in optical plasticity first became evident at about 70% of the lens radius. Thus it is unclear what the well-defined cortex-HL horizon in fish eye lenses represents in terms of lens function; if the location of the cortex-HL horizon is ubiquitous across species, it would suggest the presence of a refractive index profile (and thus lens focusing properties) which is common across disparate taxa. The objective of this study was to test for differences in the relative location of the cortex-HL horizon across multiple taxa of teleosts and elasmobranchs, thus allowing inferences to be made about the importance of relative HL lens size in the light focusing properties of the fish eye lens.

## Materials and methods

All fish samples were collected on the spring survey of the Icelandic Marine Research Institute (MRI) off the west coast of Iceland (N 65.0–67.2 W 22.2–27.2), between 1–7 March 2021. Cod (*Gadus morhua*) (n = 30), haddock (*Melanogrammus aeglefinus*) (n = 33), thorny skate (*Amblyraja radiata*) (n = 31) and round ray (*Rajella fyllae*) (n = 11) were caught in 17 bottom trawls at depths ranging from 173–308 m and bottom temperatures of -0.1 to 5.9˚C. Fish were

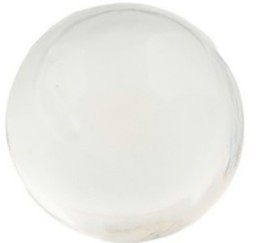
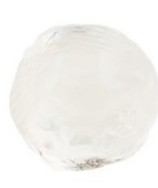

1 cm

**Fig 1. Eye lens (left side) and hardened lens (right side) of a 72-cm Atlantic cod.**

measured to the nearest 0.5 cm total length (TL) then euthanized by direct spinal cord transection. Working with the freshly-collected samples, the intact eye lens was removed with forceps through a small incision in the eye cornea and its diameter measured to the nearest 0.05 mm with calipers. The hardened portion of the eye lens (HL) was extracted from the lens by rupturing the outer membrane and rolling the lens between the fingers until an unambiguous hard central core was obtained and no additional layers could be removed (Fig 1). The same endpoint was reached if the eye was frozen, subsequently thawed, and then delaminated using forceps following the method of Wallace et al. (2014). As noted by Fernald and Wright (1983), the gelatinous outer cortex of the lens was easily distinguished from the nearly-incompressible HL; to confirm the absence of error in identifying the horizon, comparative measurements were made of the HL diameter in left and right eyes of all four fish species. As with the intact lens, HL diameter was measured with calipers to the nearest 0.05 mm. No attempt was made to identify or isolate the CHL.

All fish samples were collected on annual Icelandic federal government research surveys with the approval of the Animal Care Committee of the Marine and Freshwater Research Institute in Reykjavik.

## Results

Both lens diameter (LD) and hardened lens diameter (HD) increased significantly with the total length (TL) of the fish (Table 1; Fig 2). The relationship was linear across the length range sampled in all species, including those that were clearly juveniles. Analysis of covariance indicated that the slopes differed significantly among species (ANOVA, $p < 0.01$). Left and right lens diameters were highly correlated in all species ($r > 0.98$), as were HL diameters, indicating little measurement or preparation error.

The relationship between LD and HD was linear and highly significant in all cases (Table 2; Fig 3). The slopes relating LD to HD differed slightly but significantly between teleosts and elasmobranchs (ANOVA, $p = 0.01$) but the slopes of individual species did not show a significant difference (ANOVA, $p = 0.057$); the slope for thorny skate was the only species that may have been different from the others (Table 2; Fig 3). The mean HD:LD ratio in all species combined was 0.69 (95% CI = 0.67–0.70, N = 105). The HD:LD ratio was similar across both the teleost and elasmobranch species, with a mean ratio of 0.68 (95% CI = 0.67–0.69, N = 63) in

**Table 1. Model parameters for the relationship predicting lens diameter (LD) or hardened lens diameter (HD) from total length (TL).**

| Species | | Intercept | Slope | N | R² |
|---|---|---|---|---|---|
| **Cod** | LD | 4.034 ± 0.282 | 0.087 ± 0.004 | 30 | 0.95* |
| | HD | 2.573 ± 0.215 | 0.062 ± 0.003 | 30 | 0.94* |
| **Haddock** | LD | 3.434 ± 0.335 | 0.124 ± 0.007 | 33 | 0.91* |
| | HD | 1.863 ± 0.195 | 0.097 ± 0.004 | 33 | 0.95* |
| **Thorny skate** | LD | 0.648 ± 0.347 | 0.124 ± 0.008 | 31 | 0.89* |
| | HD | 0.012 ± 0.410 | 0.096 ± 0.009 | 31 | 0.77* |
| **Round ray** | LD | -0.149 ± 0.783 | 0.144 ± 0.019 | 11 | 0.86* |
| | HD | -0.231 ± 0.675 | 0.112 ± 0.016 | 11 | 0.84* |

Estimates are ± 1 SE and significant relationships are indicated with *.

the teleost species and a mean of 0.70 (95% CI = 0.67–0.73, N = 42) in the elasmobranch species. There was little evidence of a change in the HD:LD ratio with increasing fish length, with the possible exception of haddock (p = 0.09).

## Discussion

The eye lens diameter was isometric with fish length, as has previously been observed in other fish species [1]. The fact that the lens HL diameter was also isometric with fish length has not been reported previously, but would be mathematically predictable from the relationships reported by Fernald and Wright (1983) [14]. Isometric growth of the hardened portion of the eye lens shows that the HL increases in size to keep pace with the size of the fish, consistent with the allometric relationships of many other body parts [16].

A fixed ratio between hardened eye lens diameter and overall eye lens diameter was found across a broad size range in both teleost and elasmobranch species, with a common HD:LD

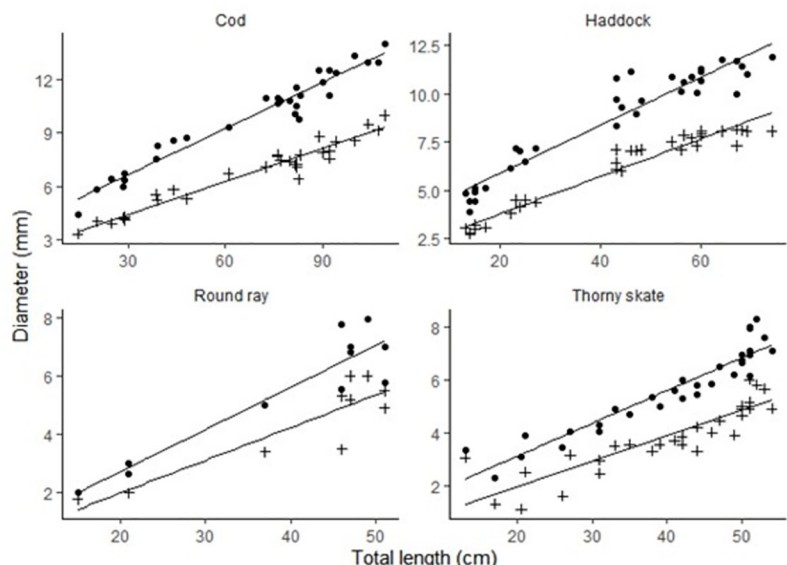

**Fig 2. The relationship between eye lens diameter (LD •), hardened lens diameter (HD +) and fish total length (TL) in two teleost and two elasmobranch species.**

**Table 2. Model parameters for the relationship predicting lens diameter (LD) from hardened lens diameter (HD).**

| Species | Intercept | Slope | N | $R^2$ | HD:LD ratio |
|---|---|---|---|---|---|
| **Cod** | 0.628 ± 0.369 | 1.379 ± 0.052 | 30 | 0.96* | 0.679 (0.66–0.69) |
| **Haddock** | 1.022 ± 0.268 | 1.282 ± 0.042 | 33 | 0.98* | 0.680 (0.66–0.69) |
| **Thorny skate** | 1.145 ± 0.250 | 1.151 ± 0.061 | 31 | 0.96* | 0.681 (0.64–0.71) |
| **Round ray** | 0.422 ± 0.501 | 1.224 ± 0.113 | 11 | 0.96* | 0.751 (0.70–0.81) |

Estimates are 1 ± SE, significant relationships are indicated with * and 95% confidence interval estimates are shown in parentheses.

ratio of 0.69. Similar observations were made in a study of the teleost *Haplochromis burtoni*, where the HL radius was 0.674 (s.d. = 0.051, N = 40) of the whole lens radius across a broad size range of fish [14]. Although the relationship between overall lens diameter and fish length was very different across taxa examined in our study, a ubiquitous and fixed HD:LD ratio shows that the growth of the HL is not only closely linked to the growth of the eye lens as a whole, but suggests that it serves a functional role that is common to disparate taxa. Fernald and Wright (1983) [14] suggested that the maintenance of optical quality during eye lens growth would require a fixed HD:LD ratio, and our findings support that hypothesis.

Most vertebrate eye lenses have a steep refractive gradient that decreases from the center of the lens towards the outer surface. A refraction gradient reduces spherical aberration and increases the total refractive power of the lens [14]. A refractive gradient in the eye lens is especially important in spherical eye lenses, such as those of teleost fishes, since a spherical eye lens holds all of the dioptric power. This is due to the very similar refractive indices of the surrounding materials, such as water, the cornea and the intraocular vitreous humour [17]. Fernald and Wright (1983) [14] measured the refractive index of the fish eye lens by measuring the path of a laser through the lens, concluding that the hardened eye lens had a uniform refractive index and that a refraction index gradient existed only in the lens cortex. They suggested that during growth, the optical qualities of the fish eye lens were preserved by

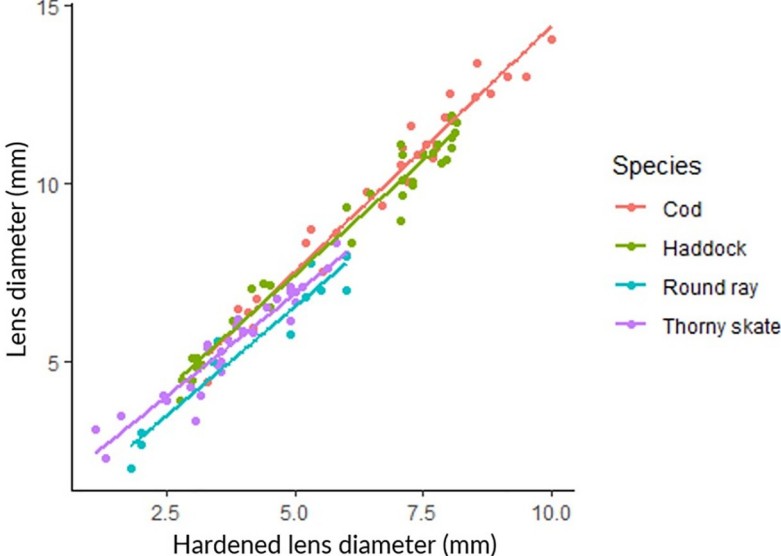

**Fig 3. The relationship between eye lens diameter (LD) and hardened lens diameter (HD) in two teleost and two elasmobranch species.**

maintaining a fixed ratio between the uniform refractive index core and the cortex containing the refraction index gradient. Fernald and Wright's interpretation has since been disputed, with Kröger (2013) [12] demonstrating the presence of a continuous refractive index gradient in the lens, with a high refractive index at the center, gradually decreasing towards the lens surface. When normalized to the lens radius, the refractive index profile remained unchanged throughout the fish's lifetime [18]. This would suggest that optical properties are maintained during growth by maintaining the gradient profile normalized to the lens radius. In a more recent study looking at a wider growth range of zebrafish (*Danio rerio*), from larva to adult, Wang et al. (2020) [20] utilized X-ray Talbot interferometry to measure the three-dimensional gradient index profiles in the eye lensemonstrating that lenses of all ages had a continuous and increasing refractive gradient from the surface towards a plateau near the center of the lens. The refractive index gradient profile changed with lens growth; when normalized to the lens radius, the slope of the profile became steeper with age, while the extent of the plateau increased with lens size and age [19]. The results of Wang et al. (2020) [20] are markedly similar to those of Fernand and Wright's (1983) [14], in that both reported a uniform refractive index at the center of the fish eye lens that increased in size with growth of the fish. Whether or not the HL represents the area of uniform refractive index is unclear. However, our results clearly showed a marked similarity in HD:LD ratios across both teleost and elasmobranch species, thus supporting the thesis that continued and precisely-maintained formation of the hardened portion of the eye lens is a necessary feature to maintain optical quality. However, our results could not be used to address the issue of the refractive index gradient profile.

Vertebrate eye lens growth begins during embryonic development, when the ectoderm overlying the optic cup inverts and pinches off to form a hollow vesicle. Cells from the posterior region of the lens elongate to form primary lens fiber cells that fill the vesicle. Newly formed cells elongate to form secondary fiber cells that overlay the primary fibers [8]. Growth of the eye lens continues throughout the lifetime of individuals with new layers constantly added on top of older layers [8, 10]. During embryonic development, fiber cells medial to the lens cortex lose their organelles and nuclei in a process resembling apoptosis, resulting in the formation of an organelle-free zone (OFZ). The primary fiber cells are the first to become organelle free but as development proceeds, more and more secondary fiber cells are included [19]. The loss of organelles allows the eye lens to achieve transparency but also hinders cells from synthesizing or degrading proteins; therefore proteins persist in the eye lens throughout the lifetime of individuals [8]. To achieve and maintain a refractive index gradient throughout the whole lens, higher protein density must exist at the center of the lens. Therefore as the lens grows, denucleated, organelle-free fiber cells must increase their protein concentration in the cytoplasm to increase their refractive index. Since denucleated, organelle-free cells are unable to synthesize protein, and since growth of the eye lens continues throughout the lifetime of the fish, compaction would appear to be necessary to increase the refractive index. Observations of flattened cortical cells in the fish eye lens could be explained by compaction [20]. However, Kozlowski and Kröger (2019) [21] developed a method for viewing cross-sections and measuring cell dimensions in fish eye lenses and demonstrated in later research consistent fiber cell thickness throughout the radius of the zebrafish eye lens, concluding that protein concentration in denucleated cells was increased by transport of proteins, likely in exchange with water, from synthetically competent cells in the periphery of the lens [22]. Schartau et al. (2009) [13] noted that lens focusing remained plastic at lens radii greater than 70%, which is the same radius threshold measured for spherical aberration in laser focusing experiments [12], and very similar to the 67% HD:LD horizon reported here. However, the OFZ occurs at 92% of the lens radius in many fish species [13]. Therefore, our results are consistent with the view that the optical plasticity of the lens is better reflected by the HD:LD horizon than by the OFZ, but

our findings do not address the mechanism for doing so. The fact that one specific refractive index (and presumably hardness and protein concentration) consistently occurred at the same relative position in different species suggests that they must have had very similar refractive index profiles.

The linear relationship between the diameter of the eye lens and the length of the fish implies that growth back-calculations of previous fish size from the lens radius should be possible [23]. Therefore, the relative size of the HL central core and laminae should be useful as indicators of the fish size corresponding to the time of lamina/core formation, even if they cannot be linked to an exact age of formation. Many recent applications of the eye lens have used the stable isotope or radiocarbon composition of the HL laminae or central core as proxies of an earlier stage of life. For example, recent applications using central cores of a very small size are currently (and reasonably) assumed to represent a very early life stage. Using standard growth backcalculation methods [23], it should be possible to estimate quite accurately the size of the fish corresponding to the size of the central core which was isolated. The only prerequisite of such an approach is the development of a predictive regression relating the length of the fish to the diameter of the hardened portion of the lens (not the lens as a whole), as was done in this and other studies [i.e. 1].

## Supporting information

**S1 Table. All corresponding data.**
(CSV)

## Acknowledgments

We are grateful for the assistance and advice provided by Jón Sólmundsson and Klara Jakobsdóttir from the Marine and Freshwater Institute of Iceland. Tomasz M. Kozłowski provided many helpful suggestions in his review of the MS.

## Author Contributions

**Conceptualization:** Steven E. Campana.

**Data curation:** Rannveig Rögn Leifsdóttir.

**Investigation:** Rannveig Rögn Leifsdóttir.

**Supervision:** Steven E. Campana.

**Writing – original draft:** Rannveig Rögn Leifsdóttir.

**Writing – review & editing:** Steven E. Campana.

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
