## [Decision Letter · Decision Letter 0]

31 Oct 2022

PONE-D-22-23795

Species independence of eye lens dimensions in teleost and elasmobranchs

PLOS ONE

Dear Dr. Leifsdóttir,

Thank you for submitting your manuscript to PLOS ONE. After careful consideration, we feel that it has merit but does not fully meet PLOS ONE’s publication criteria as it currently stands. Therefore, we invite you to submit a revised version of the manuscript that addresses the points raised during the review process.

We look forward to receiving your revised manuscript.

Kind regards,

Athanassios C. Tsikliras

Academic Editor

PLOS ONE

Journal Requirements:

Additional Editor Comments:

Specifically, the study objectives should be clearly defined in Introduction along with a clarification of the hypotheses that were examined, statistical analyses are required in the results to support the findings of the paper (also to be described in materials and methods), some parts of the discussion should be re-written and references updated and the conclusions should be formulated to comply with the findings of the study. 

Reviewers' comments:

Reviewer's Responses to Questions

**Comments to the Author**

1. Is the manuscript technically sound, and do the data support the conclusions?

Reviewer #1: Partly

2. Has the statistical analysis been performed appropriately and rigorously? 

Reviewer #1: Yes

3. Have the authors made all data underlying the findings in their manuscript fully available?

Reviewer #1: Yes

4. Is the manuscript presented in an intelligible fashion and written in standard English?

Reviewer #1: Yes

5. Review Comments to the Author

Reviewer #1: ### Introduction

The authors start with a summary of latest interest regarding vertebrate eye lenses, after which they outline the lens development and an overall structure of a lens. They expand that topic describing three, significant to their research, regions of a fish eye lens: outer cortex, hard core, and embryonic region. This section could use a clarification. On the first pass I misunderstood that the lens capsule was one of the three regions. I suggest either 1) enumerating numerically or textually the regions as they are listed in text (as in this very line) or 2) listing all three regions before discussing them. The authors describe an issue with inconsistency regarding naming in the literature, defining clearly what terms will they use for each region of interest (Hardened lens and central hardened lens, HL and CHL respectively).

The next section the authors start strong claiming HL plays central role in maintaining focused vision. I cannot agree with that statement. HL’s role is as important as the rest of the lens in forming image. In the study the authors refer to (Schartau et al. 2009), the changes in optical properties happen pass the cortex-HL horizon at 70% radius (R) demonstrating that not only the HL plays an important role. The authors pose a question of what the function of such a well-defined cortex-HL horizon is. They suggest that ubiquity of horizon’s location points to optical properties and put in the opposition the transparency caused by the loss of intracellular structures. I am not sure why those two were selected as opposing hypotheses since the intracellular loss happens way pass the 70% R all the way to 92% R. This makes the work’s motivation somewhat weak. Despite that, the authors close the section with clearly defined study objectives: testing for relative location of cortex-HL horizon in different species.

### Materials and methods

The authors provide a detailed description of samples collection: time, location, conditions, number of individuals, and their species. They describe the methodology regarding euthanasia, extraction of lens from a fish eye, extraction of HL from a lens, and measurements of fishes, lenses, and HL. Materials and methods section is solid and detailed. However, “error in identifying the horizon (…) was virtually impossible” has no place in scientific publication. Errors can be highly unlikely but never impossible, especially when the used method involves squashing a gelatinous sphere with fingers. This was particularly surprising to read because the reminder of this sentence describes the best possible approach to the problem: measuring lenses from both eyes and comparing them. Which is exactly what the authors did.

### Results

Firstly, the authors report that lens diameter (LD) and HL diameter (HD) are linearly correlated with total length (TL) of the fish, but the relation is different for different species. The relation between LD and HD is also linear. There is no significant difference between individual species, only when two groups (teleost vs elasmobranchs) were compared. This is likely caused by thorny skate which stands out from the rest. The authors establish combined for all species HD:LD ratio to be 69%. They also provide a breakdown of ratios for both groups (68% and 70%). The ratio is also invariant to TL. The authors back all their findings with statistical analysis. At the end the authors include data from five lenses of species not involved in this study. They find no correlation between DH and fork length, but they point out small sample size and sample size range. This part brings nothing to the study and makes me wonder why mentioning this at all.

### Discussion

The authors discuss that the link between LD and HL agrees with other studies and that HL increases its size to keep the pace with the fish size. The ratio of HD:LD is 69% which also agrees with reports on other species. The authors argue that whereas the relation between TL and LD is different for different species, consistent HD:LD ratio suggest HL increasing size is linked to maintenance of optical quality during the lens growth. Further the authors explain that refractive index gradient (decreasing from the center to the surface) present in most vertebrate eyes increases the optical power and reduces spherical aberration. The refractive index gradient is especially important to spherical lenses (such as in fishes) due to surrounding materials of similar refractive index (water, cornea, intraocular vitreous humor).

Unfortunately, in the next part the authors bring up outdated views regarding the refractive index profile in a fish eye lens. They refer to conclusions drawn by Fernald and Wright (1983) that the HL has a uniform refractive index, that optical properties are maintained during the growth by maintaining the cortex-HL horizon, and that central fibers are compressed due to high refractive index. Continuous gradient of refractive index has been demonstrated in Kröger 2013. A study that involved Fernald himself (Kröger et al. 2001, DOI: 10.1016/s0042-6989(00)00283-2) showed that optical properties are maintained during the growth by maintaining the gradient profile normalized to radius. The compression of fibers has also been disproved relatively recently (Kozłowski and Kröger 2019, DOI: 10.1016/j.visres.2019.06.008). The authors comment on Fernald and Wright’s conclusions by mentioning early studies that disagreed with Fernald and Wright but, as publications I brought up, demonstrate it is far from a debate. The section is closed with a conclusion that HD:LD ratio similar between species is important for maintenance of optical quality.

The authors continue with a summary of lens development leading to the fact that fiber cells lose their organelles and nuclei. They also mention that cells lose the water which is not the case. The authors seem surprised that organelle free zone (OFZ) is reported up to 92% R whereas the HL is only up to 70% R. Presence of water is exactly the reasons. It would be impossible to rub off the gelatinous outer cortex if loss of organelles would go with a loss of water. The authors also believe that 70% R reported by Schartau et al. 2009 is a threshold of optical plasticity. In Schartau’s work one can observe small differences at regions as close as 20% R (Day vs Night). If those differences are too small to be convincing, Jönsson et al. 2014 (DOI: 10.1007/s00359-014-0941-z) showed changes in lenes of Atlantic cod along almost the entire radius below the 70% R. Naturally, based on those wrong assumptions the authors build an incorrect conclusion that optical plasticity is better reflected by the HL rather than by the OFZ, where studies shows neither seem to really affect it.

The discussion is closed with a possible application of back calculations for estimates of fish sizes based on the size of isolated central core.

### Final remarks

I know my review will not make the authors’ day particularly good, but I believe that the experiment was done and reported well, and the findings can contribute to our collective knowledge on lenses. The science is good but it is not a publication material in its current form. The study needs a better motivation and conclusions drawn with better understanding of relation between optics, refractive index, and proteins.

When looking at single species, fixed cortex-HL horizon means similar density of proteins, which in turn means refractive index. The horizon must be maintained in that position throughout the growth because the entire refractive index profile must remain the same. It is very interesting that one specific refractive index (thus protein concentration, thus hardness) always ends up in the same relative position for different species, suggesting that they must have a very similar refractive index profile.

6. PLOS authors have the option to publish the peer review history of their article (what does this mean?). If published, this will include your full peer review and any attached files.

Reviewer #1: **Yes: **Tomasz M. Kozłowski

---

## [Author Response · Author response to Decision Letter 0]

2 Feb 2023

All comments have been addressed in the rebuttal letter named "response to reviewers".

---

## [Decision Letter · Decision Letter 1]

27 Mar 2023

PONE-D-22-23795R1Species independence of eye lens dimensions in teleost and elasmobranchsPLOS ONE

Dear Dr. Leifsdóttir,

Thank you for submitting your manuscript to PLOS ONE. After careful consideration, we feel that it has merit but does not fully meet PLOS ONE’s publication criteria as it currently stands. Therefore, we invite you to submit a revised version of the manuscript that addresses the three minor points raised during the review process.

We look forward to receiving your revised manuscript.

Kind regards,

Athanassios C. Tsikliras

Academic Editor

PLOS ONE

Journal Requirements:

Reviewers' comments:

Reviewer's Responses to Questions

**Comments to the Author**

1. If the authors have adequately addressed your comments raised in a previous round of review and you feel that this manuscript is now acceptable for publication, you may indicate that here to bypass the “Comments to the Author” section, enter your conflict of interest statement in the “Confidential to Editor” section, and submit your "Accept" recommendation.

Reviewer #1: (No Response)

2. Is the manuscript technically sound, and do the data support the conclusions?

Reviewer #1: Yes

3. Has the statistical analysis been performed appropriately and rigorously? 

Reviewer #1: Yes

4. Have the authors made all data underlying the findings in their manuscript fully available?

Reviewer #1: Yes

5. Is the manuscript presented in an intelligible fashion and written in standard English?

Reviewer #1: Yes

6. Review Comments to the Author

Reviewer #1: The authors addressed all the concerns from the original review. I am returning with one minor remark, few mistakes, and a comment. I will discuss them as they appear in text:

Line 79: "...a refractive index profile (...) which are common" - I think it should be "is"

Lines 167-175: The way it is formulated suggests a direct contradiction. Both Kröger and Wang talk about optical properties with lens growth - constant vs changing, but both mean the "lens growth" in a different way. Wang looked at the full growth from larva to adult, whereas Kröger looked at individuals form sexual maturity up to 3 years. Wang's results for similar range show that the changes are very small (117dpf - 880dpf). I leave the clarification at the authors' discretion.

Lines 172-173: "...continuous and declining refractive gradient from the surface towards a plateau near the center of the lens" - Refractive index is the highest in the center, so it should be "increasing" instead of "declining". Alternatively: "...declining refractive gradient from a plateau near the center of the lens towards the surface"

Line 206: "likely in exchange with water" - This part can be removed. The mechanism we suspected was transport of proteins which would dissolve in water rather than exchange it.

Line 177: "both reported a uniform refractive index at the center of the fish eye lens" - I have been thinking about this for a very long time. I disagree with this statement but I have trouble accusing the authors of being incorrect. To me Wang's plateau is not a region of constant refractive index, but rather a region in which the change is very small. However, what does it mean "constant"? What differentiate "constant" from "plateau" by my definition, would be a difference in refractive index several places after comma. Others may look at this with different precision or threshold. As such it is more of an opinion rather than a fact, making it an academic discussion which is outside the scope of a review. Nevertheless I am bringing authors' attention to this detail.

7. PLOS authors have the option to publish the peer review history of their article (what does this mean?). If published, this will include your full peer review and any attached files.

Reviewer #1: **Yes: **Tomasz M Kozłowski

---

## [Author Response · Author response to Decision Letter 1]

11 May 2023

Response to Reviewers of PONE-D-22-23795R1 - Species independence of eye lens dimensions in teleost and elasmobranchs

We thank the reviewer for his helpful comments. In the following, we have put in the reviewer’s comments and our answers marked with (AS:).

Reviewer #1: The authors addressed all the concerns from the original review. I am returning with one minor remark, few mistakes, and a comment. I will discuss them as they appear in text:

Line 79: "...a refractive index profile (...) which are common" - I think it should be "is"

AS: The wording has been changed, “are” has been corrected to “is”. 

Lines 167-175: The way it is formulated suggests a direct contradiction. Both Kröger and Wang talk about optical properties with lens growth - constant vs changing, but both mean the "lens growth" in a different way. Wang looked at the full growth from larva to adult, whereas Kröger looked at individuals form sexual maturity up to 3 years. Wang's results for similar range show that the changes are very small (117dpf - 880dpf). I leave the clarification at the authors' discretion.

AS: Agree, we have changed the wording slightly so that it does not suggest a direct contradiction. 

Lines 172-173: "...continuous and declining refractive gradient from the surface towards a plateau near the center of the lens" - Refractive index is the highest in the center, so it should be "increasing" instead of "declining". Alternatively: "...declining refractive gradient from a plateau near the center of the lens towards the surface"

AS: The wording has been changed, “declining” has been corrected to “increasing”.

Line 206: "likely in exchange with water" - This part can be removed. The mechanism we suspected was transport of proteins which would dissolve in water rather than exchange it.

AS: The part “likely in exchange with water” has been removed.

Line 177: "both reported a uniform refractive index at the center of the fish eye lens" - I have been thinking about this for a very long time. I disagree with this statement but I have trouble accusing the authors of being incorrect. To me Wang's plateau is not a region of constant refractive index, but rather a region in which the change is very small. However, what does it mean "constant"? What differentiate "constant" from "plateau" by my definition, would be a difference in refractive index several places after comma. Others may look at this with different precision or threshold. As such it is more of an opinion rather than a fact, making it an academic discussion which is outside the scope of a review. Nevertheless I am bringing authors' attention to this detail.

AS: We agree with the reviewer that the difference of “constant” and “plateau” could be an academic discussion. However, we have a different understanding of Wang’s plateau and would like to keep the original wording.

---

## [Editor Report · Decision Letter 2]

16 May 2023

Species independence of eye lens dimensions in teleost and elasmobranchs

PONE-D-22-23795R2

Dear Dr. Leifsdóttir,

We’re pleased to inform you that your manuscript has been judged scientifically suitable for publication and will be formally accepted for publication once it meets all outstanding technical requirements.

Kind regards,

Athanassios C. Tsikliras

Academic Editor

PLOS ONE
---

## [Editor Report · Acceptance letter]

22 May 2023

PONE-D-22-23795R2 

Species independence of eye lens dimensions in teleosts and elasmobranchs 

Dear Dr. Leifsdóttir:

I'm pleased to inform you that your manuscript has been deemed suitable for publication in PLOS ONE. Congratulations! Your manuscript is now with our production department. 

Kind regards, 

on behalf of

Professor Athanassios C. Tsikliras 

Academic Editor

PLOS ONE